# Using machine learning and single nucleotide polymorphisms for improving rheumatoid arthritis risk Prediction in postmenopausal women

Yingke Xu[1,2], Qing Wu [3]*

1 Nevada Institute of Personalized Medicine, College of Science, University of Nevada, Las Vegas, Nevada, United States of America, 2 Department of Epidemiology and Biostatistics, School of Public Health, the University of Nevada Las Vegas, Las Vegas, Nevada, United States of America, 3 Department of Biomedical Informatics, College of Medicine, The Ohio State University, Columbus, Ohio, United States of America

* qing.wu@osumc.edu

## Abstract

Genetic factors contribute to 60-70% of the variability in rheumatoid arthritis (RA). However, few studies have used genetic variants to predict RA risk. This study aimed to enhance RA risk prediction by leveraging single nucleotide polymorphisms (SNPs) through machine-learning algorithms, utilizing Women's Health Initiative data. We developed four predictive models: 1) based on common RA risk factors, 2) model 1 incorporating polygenic risk scores (PRS) with principal components, 3) model 1 and SNPs after feature reduction, and 4) model 1 and SNPs with kernel principal component analysis. Each model was assessed using logistic regression (LR), random forest (RF), eXtreme Gradient Boosting (XGBoost), and support vector machine (SVM). Performance metrics included the area under the receiver operating characteristic curve (AUC), sensitivity, specificity, positive and negative predictive values (PPV and NPV), and F1-score. The fourth model, integrating SNPs with XGBoost, outperformed all other models. In addition, the XGBoost model that combines genomic data with conventional phenotypic predictors significantly enhanced predictive accuracy, achieving the highest AUC of 0.90 and an F1 score of 0.83. The DeLong test confirmed significant differences in AUC between this model and the others (p-values < 0.0001), particularly highlighting its efficacy in utilizing complex genetic information. These findings emphasize the advantage of combining in-depth genomic data with advanced machine learning for RA risk prediction. The most robust performance of the XGBoost model, which integrated both conventional risk factors and individual SNPs, demonstrates its potential as a tool in personalized medicine for complex diseases like RA. This approach offers a more nuanced and effective RA risk assessment strategy, underscoring the need for further studies to extend broader applications.

## Author summary

In this study, we explored the role of genetic factors, which account for 60-70% of rheumatoid arthritis (RA) risk variability, by utilizing genetic data, specifically single

**Data availability statement:** Data was obtained from the dbGap, please find the data from https://www.ncbi.nlm.nih.gov/projects/gap/cgi-bin/study.cgi?study_id=phs000200.v12.p3

**Funding:** This work is supported by the National Institute on Minority Health and Health Disparities (R21MD013681 awarded to QW), the National Institute on Aging (R01AG080017 awarded to QW), and the National Institute of General Medical Sciences (P20GM121325-8463 awarded to QW). The funders had no role in the study design, data collection and analysis, decision to publish, or preparation of the manuscript.

nucleotide polymorphisms (SNPs). Using Women's Health Initiative data, we developed four advanced machine-learning models to predict RA risk. These models ranged from integrating common RA risk factors to sophisticated SNP analysis. The most effective model employed the eXtreme Gradient Boosting (XGBoost) method, combining SNPs with conventional risk factors, significantly enhancing predictive accuracy. This model outperformed others, achieving the highest accuracy as indicated by key metrics like the area under the curve (AUC) and F-1 score. Our findings underscore the potential of integrating detailed genetic information with machine learning in predicting RA risk, marking a significant advancement in personalized medicine, especially for postmenopausal women. This approach paves the way for more tailored healthcare strategies.

## Introduction

Rheumatoid arthritis (RA) is a chronic autoimmune disease posing significant global health challenges, affecting about 1% of the global population and leading to substantial morbidity and mortality [1,2]. As of 2020, the age-standardized global prevalence rate of RA was approximately 208.8 cases per 100,000 population [3], showing an increasing trend, especially in females. This prevalence contributes to substantial work disability, affecting around 35% of individuals with RA [4], thereby imposing considerable burdens on individuals, families, and communities.

Early and accurate diagnosis of RA is critical for effective management. Yet, it remains challenging due to overlapping symptoms with other types of arthritis and a limited window for effective treatment intervention [5]. Traditional risk factors for RA, such as age, race/ethnicity, physical activity, smoking status, and body mass index (BMI), have been well-documented [6–9]. However, the integration of genetic information for RA risk prediction remains underexplored.

Advancements in genome-wide association studies (GWAS) have shed light on genetic variants associated with RA, accounting for a significant portion of the variation in RA liability [10]. Our prior research, utilizing a polygenic risk score (PRS) derived from a pruning and thresholding method, indicated a higher RA risk in individuals with a high PRS [11]. Yet, this approach, assuming linear additive effects of genetic variants, may not reflect the complexity of RA's genetic underpinnings.

In the present study, we explore the potential of machine learning (ML) – a branch of artificial intelligence that enables predictive modeling from complex datasets [12]. ML's application in medical fields like oncology has shown promise in diagnosis, recurrence, and prognosis predictions [13,14]. This study aims to harness ML's power using single nucleotide polymorphisms (SNPs) to develop a comprehensive predictive model for RA. We will compare the performance of logistic regression, random forest, eXtreme Gradient Boosting, and support vector machine algorithms in RA risk prediction against traditional phenotype or PRS-based methods. This endeavor could revolutionize RA's early diagnosis and risk stratification, providing insights into its genetic architecture and facilitating personalized treatment approaches.

By focusing on this innovative approach, the present study aligns with the digital health paradigm, aiming to contribute significantly to personalized medicine and the management of RA, particularly in postmenopausal women who represent a high-risk group. This research thus addresses a critical gap in RA diagnosis and management, situating itself at the forefront of digital health and ML applications in chronic disease management.

## Results

A total of 12,028 participants were included in this study, 1304 with and 10,724 without RA. The mean (SD) age of participants was 61.7 (7.5) and 62.5 (7.4) years in participants with and without RA, respectively, while the mean (SD) BMI was 30.6 (5.5) and 29.1 (5.4) in the two groups. Significant age and BMI differences were observed among participants with and without RA (Table 1). The distribution of race and physical activity significantly differed among the two groups, as shown in Table 1.

Fig 1 shows the AUCs of the four models with four algorithms (including LR, RF, XGBoost, and SVM). The models using individual SNPs outperformed model 1 (with phenotype) and model 2 (with PRS) in the four algorithms. The results of the DeLong test (Table 2) showed significant differences between model 3 and model 4, with the model with phenotype and PRS (models 1 & 2) in the four algorithms (all p-values ≤ 0.01). Model 3 had a better performance than model 1 and model 2, with a significantly higher AUC than its counterpart in model 1 and model 2 (DeLong test, p-value ≤ 0.01). Fig 2 presents the sensitivity, specificity, PPV, NPV, and F1-score of the four models in four algorithms. Model 3 had a higher F-1 score when compared to models 1 and 2. Model 3 with XGBoost had a better F-1 score (F1-score=0.76) among the four ML algorithms. With this model, important SNPs can be identified by variable importance, and the top ten highest-ranking important SNPs in model 4 with XGBoost are shown in Fig 3.

Fig 1 indicates that model 4 had the highest AUC among the four models. The DeLong test shows significant differences in AUC between model 4 and model 3 (all p-values<0.0001) in the three algorithms except for the RF. Fig 2 shows that model 4 had the highest F1 score of all models. Among the four algorithms, model 4 with XGBoost had a better performance, achieved the best F1-score of 0.83, and its corresponding sensitivity, specificity, PPV, and NPV were 0.73,

**Table 1. Baseline Characteristics of Participants according to Rheumatoid Arthritis Status from Three Substudies (the Women's Health Initiative Memory Study, Population Architecture using Genomics and Epidemiology, and Genomics and Randomized Trials Network) of the Women's Health Initiative Study (N=12,028).**

|  | With RA (N=1304) | Without RA (N=10,724) | p-value* |
|---|---|---|---|
| Age, mean (SD) | 61.7 (7.5) | 62.5 (7.4) | 0.0002 |
| BMI, mean (SD) | 30.6 (5.5) | 29.1 (5.4) | <0.0001 |
| Race, n (%) |  |  |  |
| Caucasian | 190 (14.6) | 3053 (28.5) | <0.0001 |
| African American | 596 (45.7) | 4349 (40.5) |  |
| Hispanic | 443 (34.0) | 2678 (25.0) |  |
| American Indian/Alaska Native | 61 (4.7) | 343 (3.2) |  |
| Asian/Pacific Islander | 14 (1.0) | 301 (2.8) |  |
| Physical activity, n (%) |  |  |  |
| 0 MET/wk | 307 (23.5) | 2268 (21.1) | 0.03 |
| <7.5 MET/wk | 466 (35.7) | 3770 (35.2) |  |
| 7.5-15 MET/wk | 276 (21.2) | 2224 (20.7) |  |
| ≥15 MET/wk | 255 (19.6) | 2462 (23.0) |  |
| Smoking status, n (%) |  |  |  |
| Never smokers | 689 (52.8) | 5719 (53.3) | 0.5 |
| Former smokers | 476 (36.5) | 3975 (37.1) |  |
| Current smokers | 139 (10.7) | 1030 (9.6) |  |

*p - values were obtained by t-test for continuous variables and chi-square test for categorical variables.

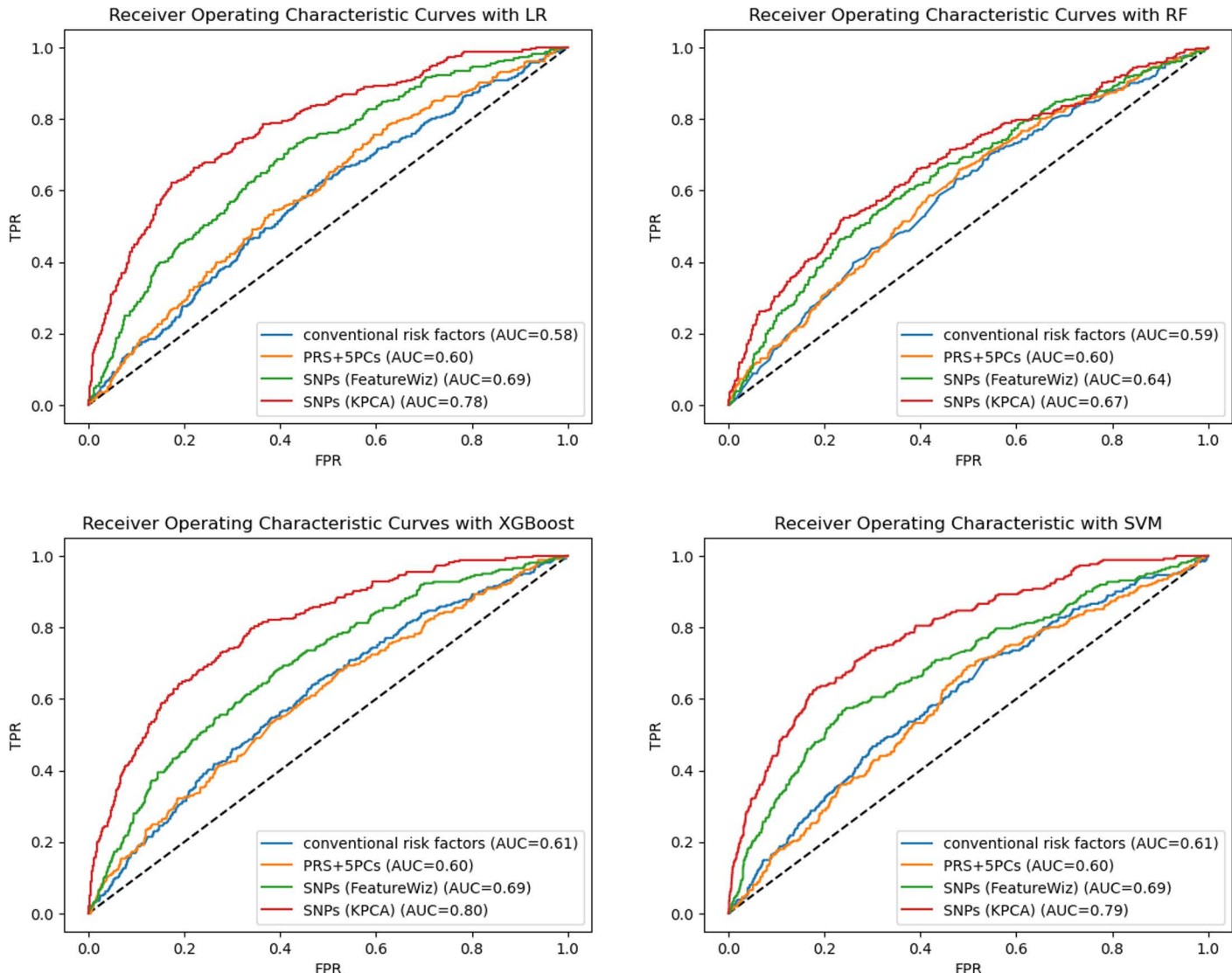

**Fig 1. Comparison of Prediction Areas Under the Curve (AUCs) on the Testing Dataset between Different Models.** *LR: logistic regression; RF: random forest; XGBoost: eXtreme Gradient Boosting; SVM: support vector machine; PRS: polygenic risk score; PCs: principal components; FeatureWiz: uses Searching for Uncorrelated List of Variables and XGBoost for feature reduction; KPCA: kernel principal component analysis; FPR: false positive rate; TPR: true positive rate.

0.72, 0.95, and 0.25, respectively. The results of AUC and other metrics indicate that XGBoost had a better performance in RA prediction when compared to LR, RF, and SVM.

Moreover, integrating conventional risk factors (Model 1) and SNP data with KPCA (Model 4) into the XGBoost model significantly improved its performance. This integrated approach achieved a notable AUC of 0.90, outperforming both Model 1 and Model 4 (p-values <0.0001). The model also demonstrated robust results with an F-1 score of 0.87, sensitivity of 0.83, specificity of 0.79, a positive predictive value (PPV) of 0.97, and a negative predictive value (NPV) of 0.33.

## Discussion

This study investigated the utility of incorporating genomic information with ML algorithms for predicting RA risk in postmenopausal women. Our findings indicate that using high

**Table 2. DeLong Tests for Comparing Area Under the Receiver Operating Characteristics Curves of Different Models with Four Algorithms in Testing Dataset.**

| | Model 1 (Phenotype information) | Model 2 (Model 1+PRS) | Model 3 (Model 1+SNPs with FeatureWiz) |
|---|---|---|---|
| **LR** | | | |
| Model 2(Model 1+PRS) | 0.4 | – | – |
| Model 3 (Model 1+SNPs with FeatureWiz) | <0.0001 | <0.0001 | – |
| Model 4(Model 1+SNPs with KPCA) | <0.0001 | <0.0001 | <0.0001 |
| **RF** | | | |
| Model 2(Model 1+PRS) | 1 | – | – |
| Model 3 (Model 1+SNPs with FeatureWiz) | 0.007 | 0.0103 | – |
| Model 4(Model 1+SNPs with KPCA) | <0.0001 | <0.0001 | 0.0562 |
| **XGBoost** | | | |
| Model 2(Model 1+PRS) | 0.7 | – | – |
| Model 3 (Model 1+SNPs with FeatureWiz) | 0.0007 | <0.0001 | – |
| Model 4(Model 1+SNPs with KPCA) | <0.0001 | <0.0001 | <0.0001 |
| **SVM** | | | |
| Model 2(Model 1+PRS) | 0.6 | – | – |
| Model 3 (Model 1+SNPs with FeatureWiz) | <0.0001 | <0.0001 | – |
| Model 4(Model 1+SNPs with KPCA) | <0.0001 | <0.0001 | <0.0001 |

LR: logistic regression; RF: random forest; XGBoost: eXtreme Gradient Boosting; SVM: support vector machine; FeatureWiz uses Searching for Uncorrelated List of Variables algorithms and XGBoost to reduce features; KPCA: kernel principal component analysis. Model 1: using phenotype information as predictors; Model 2: using polygenic risk score and five principal components as predictors+Model1; Model 3: using SNPs (FeatureWiz) information as predictors+Model1; Model 4: using SNPs (Kernel principal component analysis) information as predictors+Model1.

dimensional genomic data with ML models outperformed models using phenotype or PRS data. Additionally, XGBoost outperformed other algorithms regarding RA prediction among postmenopausal women. Moreover, integrating genomic information with traditional phenotypic predictors using the XGBoost model significantly enhanced its predictive accuracy, achieving an F-1 score of 0.87 and an AUC of 0.90.

Several GWAS have identified hundreds of SNPs associated with RA. Some studies observed that individuals with a family history of RA significantly increase the risk of developing the disease [15], indicating that using genetic variants as predictors may help predict an individual's RA risk. PRS has been used to summarize information from genetic variants as a single value for estimating the risk for a specific disease. However, PRS often have limited predictive power because the effect sizes of genetic variants are small. Thus, PRS may not adequately capture the polygenic liability of RA. In one of our previous studies, a significant association between PRS and RA was observed but with a small effect size, so the PRS might not perform well in risk prediction in practice. In contrast, treating each SNP as an individual predictor might be beneficial since PRS has an assumption about the additivity, while treating SNPs as predictors do not rely on such an assumption; thus, this method might be more robust for disease prediction in personalized medicine.

ML has been used in several studies to predict complex disease phenotypes using SNPs, including RA prediction. A recent study used electronic health records with a deep learning model to accurately predict the activity of RA [16]. While ML has been used for RA prediction in previous studies, our study is the first to utilize SNPs as predictors with ML algorithms to predict RA risk. We found that models using single SNPs as predictors performed better than models using phenotype or PRS only. Our results suggest that high dimensional genomic data with ML models can significantly improve RA risk prediction over models using phenotype

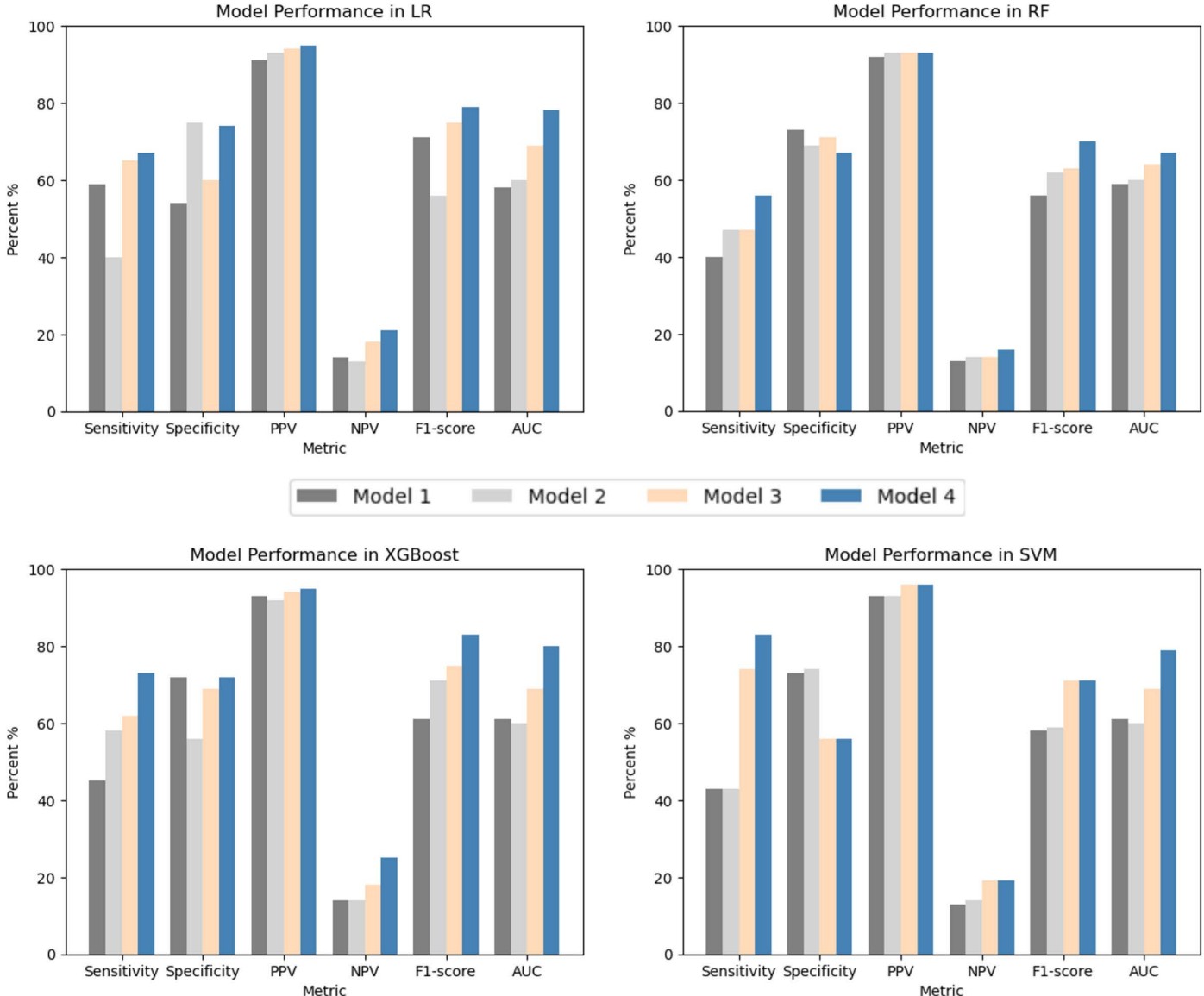

**Fig 2. Comparison of Model Performance on the Testing Dataset between Different Models.** *LR: logistic regression; RF: random forest; XGBoost: eXtreme Gradient Boosting; SVM: support vector machine; PPV: positive predictive value; NPV, negative predictive value; AUC: Areas Under the Curve.

or PRS data. ML models can potentially capture interaction effects between SNPs and other risk factors, thus capturing more information and improving RA prediction. In addition, we utilized KPCA for dimensionality reduction [17]. Although principal component analysis (PCA) is more used in practice, KPCA can handle nonlinear data while PCA can only perform linear dimensionality reduction [18]. Several studies utilized KPCA for feature reduction and observed that such a method could effectively capture the underlying structure and information [17,19]. The current study found that the KPCA model outperformed the FeatureWiz model.

Our study acknowledges several limitations. The focus on postmenopausal women in the United States may limit the broader applicability of our findings, underscoring the need for further research across diverse populations to enhance the model's generalizability in RA

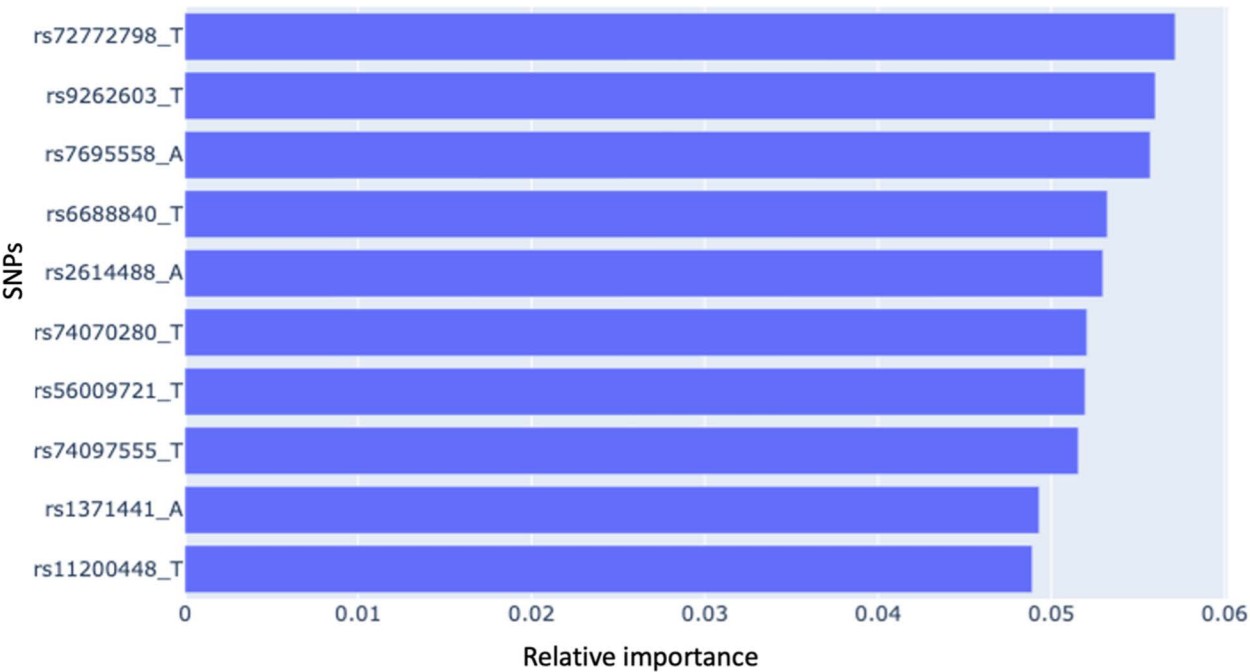

**Fig 3. Relative Feature Importance of the Top 10 SNPs in Model 3 (with single nucleotide polymorphisms after FeatureWiz) with Extreme Gradient Boosting.**

risk prediction. Additionally, due to limitations in data access, only a narrow set of phenotype information (e.g., history of live birth) was utilized in the study. Broader phenotype risk factors should be explored in future research. Moreover, using randomly split training and test datasets, though effective in mitigating overfitting, might not capture the complexities of real-world data [20]. Hence, future studies should employ cross-validation and external validation to augment the model's robustness and practical applicability. A notable limitation is the inability of our model to differentiate between RA subtypes due to the constraints of the WHI data, which is a critical aspect, considering the varied clinical manifestations and treatment responses of seropositive and seronegative RA. Future research should include subtype-specific data to develop more accurate models, paving the way for personalized risk predictions and treatment strategies. Addressing these limitations will enhance our model's predictive accuracy and clinical relevance, making it a more effective tool for personalized RA risk assessment.

Our findings highlight a significant advancement in RA risk prediction through integrating genomic information with ML algorithms, notably outperforming models based on phenotype or PRS data. XGBoost has proven to be the most effective algorithm in postmenopausal women. This research provides crucial insights into using ML and genomic data for predicting complex diseases like RA, with implications extending to clinical and public health realms. Note that using the conventional clinical risk factors and SNPs with KPCA with XGBoost substantially improves predictive performance. This finding suggests that our model is promising to enhance risk prediction in clinical scenarios, potentially offering benefits in more accurate patient stratification and personalized medicine. The improved predictive accuracy of our model offers healthcare providers a tool for more efficient identification of high-risk individuals, enabling earlier and more customized interventions. Moreover, these methods could enhance public health strategies, leading to better-targeted screening and resource allocation

for RA management. Continued research is necessary to extend these methods to a broader range of populations, ultimately enriching their applicability. The long-term value of these approaches lies in their potential to transform personalized medicine, especially in the risk assessment and management of RA.

## Materials and methods

### Design and participants

This study analyzed data from the Women's Health Initiative (WHI), a longitudinal study conducted among postmenopausal women at 40 clinical centers across the United States. The study design has been previously described [21]. Briefly, 161,808 women were recruited for one or more of three clinical trials or an observational study. Participants received physical examinations yearly or every three years and additional data were collected through question-naires via mail or telephone. Every participating institution obtained approved consent forms from the Institutional Review Board. Data for this study were obtained from the database of Genotypes and Phenotypes with the approval of the institutional review board at the University of Nevada, Las Vegas. The data was fully anonymized before we accessed them, and UNLV IRB waived the informed consent. Access to the data was granted through the database of Genotypes and Phenotypes (https://www.ncbi.nlm.nih.gov/projects/gap/cgi-bin/study.cgi?study_id=phs000200.v12.p3).

The current study included participants from three WHI sub-studies: the Women's Health Initiative Memory Study, Population Architecture using Genomics and Epidemiology, and Genomics and Randomized Trials Network. Participants were excluded if they had arthritis at baseline or participated in hormone replacement therapy or the vitamin D trial, as these factors may impact RA [22,23]. This study conforms to the Strengthening the Reporting of Observational Studies in Epidemiology (STROBE), and the STROBE checklist can be found in S1 Checklist.

### Genotype information

Genotyping was performed using the Illumina or Affymetrix 6.0 Array Set Platform. Genotype imputation was conducted on the Sanger Imputation Server using the Haplotype Reference Consortium reference panel and Positional Burrows–Wheeler Transform imputation algorithm [24]. Summary statistics from the most powerful RA-related genome-wide association studies (GWAS) were used for SNP extraction [25]. Quality control steps described by Choi et al. were performed using PLINK 1.9 [26]. SNPs with minor allele frequencies less than 0.01, a low P-value from the Hardy-Weinberg Equilibrium, or missing in a high fraction of subjects were filtered out. Pruning was performed with a window size of 200 variants, sliding across the genome with a step size of 50 variants at a time. SNPs with LD $r^2$ higher than 0.25 were filtered out, and individuals with a first or second-degree relative in the sample were removed. Each SNP was coded as AA = 0, Aa = 1, aa = 2, implying that each additional copy of the minor allele increases the risk by the same amount. Our prior study determined an optimal threshold for PRS calculation. The current study utilized the optimal threshold to compare the model's predictive performance using SNPs as predictors and the model with PRS. An individual's PRS be derived as: where $X_j$ is the number of risk alleles (0, 1, or 2) for variant j, and $\beta_j$ is the effect size of variant j, obtained from the summary statistics of GWAS. The PRS was derived based on our previous study [11]. More specifically, seven candidate scores were derived using different p-value thresholds (0.5, 0.4, 0.3, 0.2, 0.1, 0.05 and 0.001); the optimal PRS was selected based on the maximum area under the receiver operating characteristic curve (AUC) when predicting observed RA cases, and it was subsequently used for further analyses.

## Phenotype information

Well-established risk factors of RA were used in the current study, including age, race/ethnicity, physical activity, smoking status, and BMI, which were considered predictors [6–9]. Except for BMI, other information was collected by questionnaire (more details: https://www.whi.org/formList). The participants in the current study self-reported race, including Caucasian, African American, Hispanic, American Indian/Alaska Native, Asian, and American Indian. The physical activity is normally expressed in metabolic equivalent of task (MET) units, and the expenditure of energy from recreational physical activity (MET-hours/week) was calculated based on questions about the participant's usual activity exercise. The Physical Activity Guidelines for Americans 2008 suggested a minimum of 150 min/week of moderate-intensity exercise, which equals 7.5–14.9 MET-h/week. In this study, we categorized physical activity into four levels: no exercise (0); less than the guideline (0.1–7.4 MET-h/week); meeting the guideline (7.5–14.9 MET-h/week); and exceeding the guideline (≥15 MET-h/week) [7]. For the smoking status, women were classified as current, former, or never-smokers (participants who had not smoked 100 cigarettes in their lifetime). Former smokers were defined as answering 'yes' to the question, 'Have you smoked 100 cigarettes in your life?' but' no' to 'Do you smoke cigarettes now?' Current smokers were defined as answering 'yes' to both questions [9]. The BMI was calculated based on measured weight and height. Individuals were categorized as underweight (BMI<18.5), normal weight(18.5≤BMI<25), overweight (25≤BMI<30), and obese (BMI≥30) [6]. Weight was measured in kg on a balance beam scale with the participant without shoes. Height was measured in 0.1 centimeters using a wall-mounted stadiometer.

## Statistical analysis

The dataset was divided into training (80%) and testing (20%) sections for model development and performance evaluation. The dataset was imbalanced, so under-sampling was applied [27], a method that deletes majority examples from the dataset so that the numbers of examples between different classes become balanced. Hybrid feature selection methods were utilized to address the dimensionality challenge caused by including whole genome-wide SNPs as individual predictors combining different feature selections in a multi-step process [28]. LASSO, a penalized regression method [29], was utilized for SNP selection, and 4774 SNPs were retained based on their non-zero coefficient. Two methods were employed for dimension reduction; one is FeatureWiz, a quick and effective technique that uses Searching for Uncorrelated List of Variables (SULOV) algorithms and XGBoost to reduce features. Another one is kernel principal component analysis (KPCA) [30], which can construct nonlinear mappings that maximize the variance in the data. Four models were developed for RA prediction, including model 1 (using the most common RA risk factors), model 2 (model 1 with PRS and five principal components), model 3 (model 1with SNPs after feature reduction with FeatureWiz), and model 4 (model 1with SNPS after feature extraction with KPCA). Four algorithms, including LR, RF, XGBoost, and SVM, were used for each model, and five-fold cross-validation was performed with hyperparameters tuned using random and grid search. Parameters were tuned based on the model characteristics. Performance was assessed using the area under the receiver operating characteristic curve (AUC), sensitivity, specificity, positive and negative predictive values, and F1-score. The flowchart of the current study is shown in Fig 4 The DeLong nonparametric test was used to assess differences in performance between models. In the DeLong test, the z-score is calculated to assess the null hypothesis H0:AUC1=AUC2. The z-score measures the standardized difference between the two AUCs by considering their variances and covariance. The ten most important SNPs in model 4 were

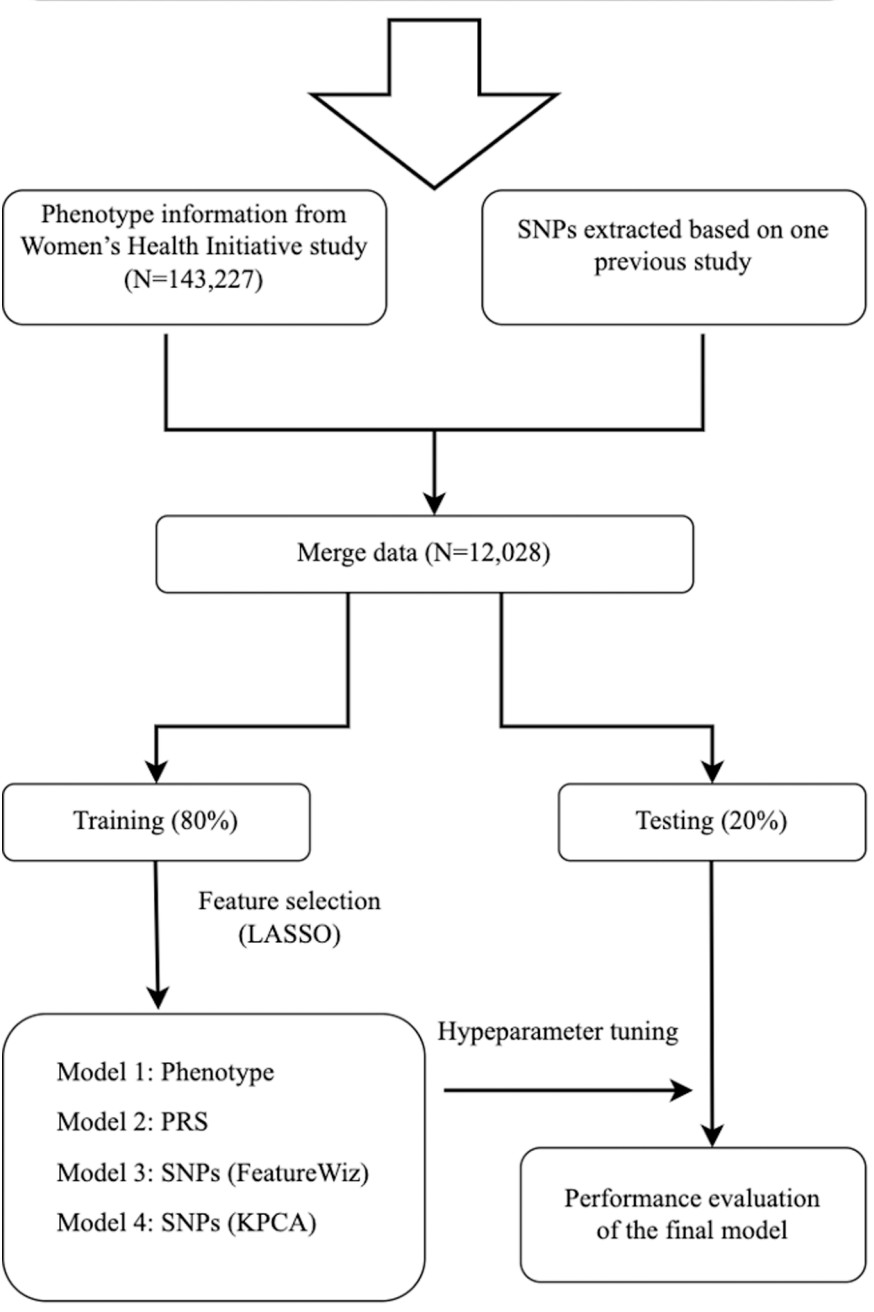

**Fig 4. Flowchart of the Study.**

evaluated [31]. The K-nearest neighbor algorithm was used for missing value imputation in phenotype information. The missing values were replaced with the mean values from similar or close neighbors, with a default number of neighbors (N=5). All statistical tests were two-sided, and the statistical significance was p < 0.05. Python programming language version 3.8.15 was used for all statistical analyses, utilizing NumPy, Pandas, imblearn, Matplotlib, Seaborn, Scikit-learn, Xgboost, and Featurewiz were used in this study [32–39].

## Supporting information

**S1 Checklist. STROBE checklist.**
(DOCX)

AcknowledgmentWe sincerely thank the original WHI study investigators and the invaluable participants for their pivotal contributions to advancing women's health research. We also express our gratitude to the National Institutes of Health (NIH) and the database of Geno-types and Phenotypes (dbGaP) for granting access to analyze the WHI data. This work reflects our independent analysis and interpretation and does not represent the views of other parties associated with the WHI study. We sincerely appreciate the collective efforts and contribu-tions of all institutions, collaborators, and teams involved in the WHI study. Part of Dr. Qing Wu's work was conducted at the University of Nevada, Las Vegas.

## Author contributions

**Conceptualization:** Yingke Xu.

**Data curation:** Yingke Xu.

**Formal analysis:** Yingke Xu.

**Investigation:** Yingke Xu.

**Methodology:** Yingke Xu.

**Resources:** Qing Wu.

**Software:** Yingke Xu.

**Supervision:** Qing Wu.

**Validation:** Yingke Xu.

**Visualization:** Yingke Xu.

**Writing – original draft:** Yingke Xu.

**Writing – review & editing:** Yingke Xu, Qing Wu.

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
