## [Decision Letter · Decision Letter 0]

2 Jul 2024

PDIG-D-24-00034

Using machine learning and single nucleotide polymorphisms for improving rheumatoid arthritis risk Prediction in postmenopausal women

PLOS Digital Health

Dear Dr. Wu,

Thank you for submitting your manuscript to PLOS Digital Health. After careful consideration, we feel that it has merit but does not fully meet PLOS Digital Health's publication criteria as it currently stands. Therefore, we invite you to submit a revised version of the manuscript that addresses the points raised during the review process.

Please submit your revised manuscript within 60 days Aug 31 2024 11:59PM. If you will need more time than this to complete your revisions, please reply to this message or contact the journal office at digitalhealth@plos.org. Please include the following items when submitting your revised manuscript:

We look forward to receiving your revised manuscript.

Kind regards,

Henry Horng-Shing Lu

Section Editor

PLOS Digital Health

Journal Requirements:

1. Please send a completed 'Competing Interests' statement, including any COIs declared by your co-authors. If you have no competing interests to declare, please state "The authors have declared that no competing interests exist". Otherwise please declare all competing interests beginning with the statement "I have read the journal's policy and the authors of this manuscript have the following competing interests:"

3. Please provide separate figure files in .tif or .eps format only and remove any figures embedded in your manuscript file. Please also ensure that all files are under our size limit of 10MB.

Additional Editor Comments (if provided):

Reviewers' comments:

Reviewer's Responses to Questions

**Comments to the Author**

1. Does this manuscript meet PLOS Digital Health’s publication criteria ? Is the manuscript technically sound, and do the data support the conclusions? The manuscript must describe methodologically and ethically rigorous research with conclusions that are appropriately drawn based on the data presented.

Reviewer #1: Yes

Reviewer #2: Partly

2. Has the statistical analysis been performed appropriately and rigorously?

Reviewer #1: N/A

Reviewer #2: No

3. Have the authors made all data underlying the findings in their manuscript fully available (please refer to the Data Availability Statement at the start of the manuscript PDF file)?

Reviewer #1: No

Reviewer #2: No

4. Is the manuscript presented in an intelligible fashion and written in standard English?

PLOS Digital Health does not copyedit accepted manuscripts, so the language in submitted articles must be clear, correct, and unambiguous. Any typographical or grammatical errors should be corrected at revision, so please note any specific errors here.

Reviewer #1: Yes

Reviewer #2: Yes

5. Review Comments to the Author

Please use the space provided to explain your answers to the questions above. You may also include additional comments for the author, including concerns about dual publication, research ethics, or publication ethics. (Please upload your review as an attachment if it exceeds 20,000 characters)

Reviewer #1: The study applies machine learning algorithms to Women's Health Initiative data, using SNPs and traditional risk factors to predict rheumatoid arthritis risk in postmenopausal women, achieving significant predictive accuracy. I have few points to raise

In Table 1, the correct figure is 40.5, not 405.

Additional insights into the prevalence of the most influential SNPs are necessary, particularly noting the absence of overlap with SNPs commonly associated with RA, as identified in a prior literature (https://www.ncbi.nlm.nih.gov/pmc/articles/PMC3030622/)

The selection of phenotype variables was narrow , neglecting other potentially relevant factors such as he history of live births 

It would be beneficial to observe the study's outcomes both before and after the resampling process.

The absence of a separate validation isn’t addresses particularly it seems that the hyperparameters were tuned on the test data.

Reviewer #2: Major Comments:

Please provide some raw data and the relevant codes. 

For verification of the results, please provide a link for the ML models developed after training alongwith a test data for verification.

Please provide details of the Python libraries and the packages used.

Please provide supplementary data highlighting the actual phenotypic features and the Polygenic Risk Scores (PRS) features used for training various ML models. Please provide details with respect to PRS data.

Please provide details regarding the self-reporting questionnaire used for data collection towards phenotypic features and methods employed to mitigate biases during the same. 

Please provide details regarding MET and other calculations used in the study.

Please provide detailed information regarding data pre-processing and Feature Engineering methods used before training the ML models.

Please provide details regarding the hyperparametric tuning performed.

Kindly provide evidence to showcase that the baseline characteristics which significantly differ between the two cohorts aren’t contributing as confounding factors towards model building using genomic information.

Kindly provide the results as a confusion matrix for the verification of the results.

Please provide the details of the methodology and the tool used for under-sampling.

Kindly provide the data as evidence towards improvement in F1 score upon integration of genomic information with traditional phenotypic predictors. 

Kindly detail about the SNP selection methodology and the tool used.

Please detail the methodology considering the reproducibility of the results.

Minor Comments:

Please provide details for DeLong Test and various model evaluation metrics used in the paper.

6. PLOS authors have the option to publish the peer review history of their article (what does this mean? ). If published, this will include your full peer review and any attached files.

**Do you want your identity to be public for this peer review?** For information about this choice, including consent withdrawal, please see our Privacy Policy .

Reviewer #1: No

Reviewer #2: No

---

## [Decision Letter · Decision Letter 1]

17 Feb 2025

Using machine learning and single nucleotide polymorphisms for improving rheumatoid arthritis risk Prediction in postmenopausal women

PDIG-D-24-00034R1

Dear Wu,

We are pleased to inform you that your manuscript 'Using machine learning and single nucleotide polymorphisms for improving rheumatoid arthritis risk Prediction in postmenopausal women' has been provisionally accepted for publication in PLOS Digital Health.

Best regards,

Henry Horng-Shing Lu

Section Editor

PLOS Digital Health

**Additional Editor Comments (if provided):**

**Reviewer Comments (if any, and for reference):**

Reviewer's Responses to Questions

**Comments to the Author**

1. If the authors have adequately addressed your comments raised in a previous round of review and you feel that this manuscript is now acceptable for publication, you may indicate that here to bypass the “Comments to the Author” section, enter your conflict of interest statement in the “Confidential to Editor” section, and submit your "Accept" recommendation.

Reviewer #3: All comments have been addressed

2. Does this manuscript meet PLOS Digital Health’s publication criteria ? Is the manuscript technically sound, and do the data support the conclusions? The manuscript must describe methodologically and ethically rigorous research with conclusions that are appropriately drawn based on the data presented.

Reviewer #3: Yes

3. Has the statistical analysis been performed appropriately and rigorously?

Reviewer #3: Yes

4. Have the authors made all data underlying the findings in their manuscript fully available (please refer to the Data Availability Statement at the start of the manuscript PDF file)?

Reviewer #3: Yes

5. Is the manuscript presented in an intelligible fashion and written in standard English?

Reviewer #3: Yes

6. Review Comments to the Author

Reviewer #3: The revision is fine; all reviewers' concerns have been addressed.

7. PLOS authors have the option to publish the peer review history of their article (what does this mean? ). If published, this will include your full peer review and any attached files.

**Do you want your identity to be public for this peer review?** For information about this choice, including consent withdrawal, please see our Privacy Policy .

Reviewer #3: No
